# Design of a Leaf-Bottom Pest Control Robot with Adaptive Chassis and Adjustable Selective Nozzle

Dongshen Li [1], Fei Gao [1], Zemin Li [1], Yutong Zhang [1], Chuang Gao [1] and Hongbo Li [2,*]

1   College of Engineering, Northeast Agricultural University, Harbin 150030, China;
    a07210240@neau.edu.cn (D.L.); a07210347@neau.edu.cn (F.G.); a07220291@neau.edu.cn (Z.L.);
    a07220042@neau.edu.cn (Y.Z.); s220702033@neau.edu.cn (C.G.)
2   College of Electrical and Information, Northeast Agricultural University, Harbin 150030, China
*   Correspondence: lihongbo@neau.edu.cn

**Abstract:** Pest control is an important guarantee for agricultural production. Pests are mostly light-avoiding and often gather on the bottom of crop leaves. However, spraying agricultural machinery mostly adopts top-down spraying, which suffers from low pesticide utilization and poor insect removal effect. Therefore, the upward spraying mode and intelligent nozzle have gradually become the research hotspot of precision agriculture. This paper designs a leaf-bottom pest control robot with adaptive chassis and adjustable selective nozzle. Firstly, the adaptive chassis is designed based on the MacPherson suspension, which uses shock absorption to drive the track to swing within a 30° angle. Secondly, a new type of cone angle adjustable selective nozzle was developed, which achieves adaptive selective precision spraying under visual guidance. Then, based on a convolutional block attention module (CBAM), the multi-CBAM-YOLOv5s network model was improved to achieve a 70% recognition rate of leaf-bottom spotted bad point in video streams. Finally, functional tests of the adaptive chassis and the adjustable selective spraying system were conducted. The data indicate that the adaptive chassis can adapt to diverse single-ridge requirements of soybeans and corn while protecting the ridge slopes. The selective spraying system achieves 70% precision in pesticide application, greatly reducing the use of pesticides. The scheme explores a ridge-friendly leaf-bottom pest control plan, providing a technical reference for improving spraying effect, reducing pesticide usage, and mitigating environmental pollution.

**Keywords:** leaf-bottom pest control; adaptive chassis; adjustable selective nozzle; small target detection; attention mechanism

## 1. Introduction

By the middle of this century, the world will need to produce 70% to 100% more food to feed the growing population. The scarcity of arable farmland means that sustainability and scientific precision of crop protection are important trends in future agricultural development [1]. The Food and Agriculture Organization has noted that annual global field crop losses of 50% to 80% are due to pests and diseases [2]. Therefore, pest and disease control in fields plays a crucial role in maintaining and improving crop yields [3]. It is estimated that a considerable amount of crop loss is recuperated through pesticide application. Although the importance of pest and disease control is increasingly recognized, the residual effects of agricultural pests and environmental pollution threats are often overlooked [4–6]. Efforts are being made to achieve the massive increases in yields needed by the global population without further destroying natural habitats and polluting the air and water. Automated precision pesticide application is a trend to improve pesticide utilization and soil safety [7].

Current market-available pesticide application equipment includes unmanned aerial vehicle (UAV) spraying [8], traction truss spraying [9], artificial backpack spraying and

high-clearance self-propelled sprayers [10,11]. The spray droplets of these forms mainly cover the adaxial surface of the leaves. Due to the insect's avoidance of light, pests and diseases are mostly distributed on the bottom of the leaves, making the control effect limited. In order to solve the above problems, improving the pesticide application rate on the bottom of leaves has become a concern of researchers. Most of the spraying schemes mainly focus on the amount of liquid, atomization of spraying [12,13] and electrostatic spraying [14–16] to improve the spraying rate of the leaf bottom. However, most of the existing agricultural machinery is prone-spraying machinery, creating a problem of a low upper limit of the pesticide rate sprayed at the bottom of the leaves. Designing new schemes and configurations that support lateral or upward spraying is an effective attempt to increase the upper limits of leaf-bottom medication rates [17–19].

Extensive and in-depth research was conducted domestically and internationally, achieving a series of outcomes. The following solutions were proposed. Luciano Cantelli and his team [20] utilized an extended Kalman filter to fuse multi-sensor data to evaluate the robot's pose and used positioning nodes for autonomous navigation. They operated hydraulic elements through command sets and used sensors for feedback in a closed-loop control for local spraying, achieving intelligent precision spraying. Longzhe Quan et al. [21] designed a spherical robot equipped with binocular stereo vision. By shifting the center of gravity to control the posture and using a two-axis stabilization method to improve the quality of binocular recognition, they explored a new micro-scale scheme for collecting and monitoring greenhouse plant physiology. Roberto Oberti et al. [22] conducted in-depth research on selective spraying, using a six-degree-of-freedom (dof) robotic arm for selective spraying. Equipped with an integrated pest sensing system based on R–G–NIR (red, green, near-infrared) multispectral image technology, the robotic arm combined with a precision spraying end effector effectively reduced pesticide usage. Seol, Kim and Son [23] proposed a real-time variable flow control system based on deep learning for segmenting spraying areas. By optimizing the pulse width modulation (PWM) controller, they achieved intelligent spraying in orchards. The improvements in the current new solution are not comprehensive, for example, they only use existing commercial micro-chassis or lack the versatility to carry out various operations such as spraying or collection. Existing precision spraying is achieved through vehicle motion for lateral alignment or by switching array nozzles for target alignment. While these were effective, there remain some shortcomings. The main focus is on three points: (1) existing chassis schemes cannot balance efficiency and low spatial occupancy; (2) selective alignment through vehicle movement or array nozzles has low precision, making it difficult to improve pesticide utilization; and (3) most schemes are still limited to a single crop type, unable to accommodate different ridge spacings and ground clearance compatibility.

Identification of leaf spots and pests is the basis and prerequisite for precise pesticide application. Some scholars have developed a series of target detection technologies by extracting the features of the detection objects and combining common neural network models such as Region-Convolutional Neural Network (R-CNN), Residual Neural Network (ResNet) [24], and YOLO [25,26]. This has accelerated the intelligent process in the field of pest control. Although existing target detection technologies can detect pest spots, there is still much room for improvement in recognition speed and accuracy. Exploring new neural network architectures and designing high-precision recognition solutions are important ways to expand the field of crop pest control.

In response to these issues, extensive and deep research was carried out both domestically and internationally, achieving a series of results. Wan-jie Liang, Hong Zhang et al. [27] demonstrated the superiority of CNNs in recognizing rice blast disease, providing a new method for the automatic recognition and classification of rice blast disease. The results showed that this method achieved an accuracy rate of over 95%. Dandan Wang and Dongjian [28] used a channel pruning algorithm to simplify a trained YOLOv5s model, improving its efficiency while maintaining detection accuracy. The recognition accuracy can reach 95% with an average detection time of 8 ms per image. Jiangtao Qi and colleagues [29]

used an improved SE-YOLOv5 network model to extract key features from tomato disease images, achieving an accuracy of 91.07%. Jiawei Li et al. [30] proposed a combination of the CSP structure and Transformer encoder based on the YOLOv5 network to enhance the network's ability to capture disease characteristics. By using an improved InceptionA module, they could better extract features of plant diseases and pests. Chongke Bi and colleagues [31] adopted the lightweight Convolutional Neural Network MobileNet as the model for apple leaf disease identification, with a processing speed of only 0.22 seconds per image. Although computational costs increased, significant improvements were achieved in detection capabilities and global feature extraction. Most studies can accurately identify the disease and insect pest areas on leaves but the applicability of their model performance is small [32]. For example, only leaves of a single type of crop are selected, and only data with obvious disease and insect pest spots are trained. Although a high recognition accuracy rate can be achieved, the recognition system is only applicable to experimental fields. Studies have found that most crop pests, such as corn borers and potato beetles, exhibit phototaxis and thigmotaxis and are distributed on the bottom of leaves [33]. However, most research did not consider the growth habits of crop pests and only selected the front of leaves as the dataset, not meeting the actual field pest control requirements. Our primary goal is to ensure recognition accuracy and speed while conforming to real field requirements, thus improving the generalization ability of leaf-bottom spotted bad point detection.

In response to these problems, in order to meet the requirements for precise control of leaf-bottom pests, a field management robot system equipped with an adaptive chassis and adjustable selective nozzle is proposed. In Section 2, this paper carries out three contributions, namely: (1) an adaptive chassis is designed to address the problem of poor adaptability to different ridge spacings and slopes; (2) adjustable selective spraying is designed to meet the working needs of crops with different heights above the ground and scattered leaf-bottom spotted bad point areas; and (3) multi-CBAM-YOLOv5s is designed for target identification of dead spots caused by minor plant diseases and insect pests on leaves during the precise application of pesticides. Section 3 details the experiments of the adaptive chassis motion function, pest identification and spraying system to verify the effectiveness of the scheme. Section 4 concludes the paper.

The main contributions of this paper are as follows:

(1) A leaf-bottom pest control robot is designed for multiple types of ridges and different ground clearances. It achieves its goals through an adaptive chassis and adjustable selective nozzle;
(2) A multi-CBAM-YOLOv5s network is developed. It improves the accuracy of identifying tiny leaf-bottom spotted bad point defects on the leaf underside by up to 85%.
(3) A new complex multi-type agricultural scene precision selective pesticide automation scheme is provided.

## 2. Design of a Leaf-Bottom Pest Control Robot

### 2.1. General Introduction

The main body of the robot includes an adaptive chassis and an adjustable selective spraying system, as shown in Figure 1. The single-ridge walking mode was coupled with an adaptive module. An adaptive chassis not only applies to different ridge shapes but also facilitates compatibility with crops of different ground clearances. This form lays the groundwork for cluster operations. The mechanical design of the robot is primarily composed of eight main units: adjustable nozzle, two-dof platform, motor, adaptive module, battery packs, crawler, pesticide box, and control unit. The battery packs, crawler, motor, and adaptive module are symmetrically positioned on both sides of the body.

The overall design encompasses two core tasks: (1) an adaptive chassis that can dynamically adjust the opening swing angle to enter ridges without reserved driving space and (2) an adjustable selective nozzle that detects pest congregation areas and automatically adjusts parameters for pesticide application.

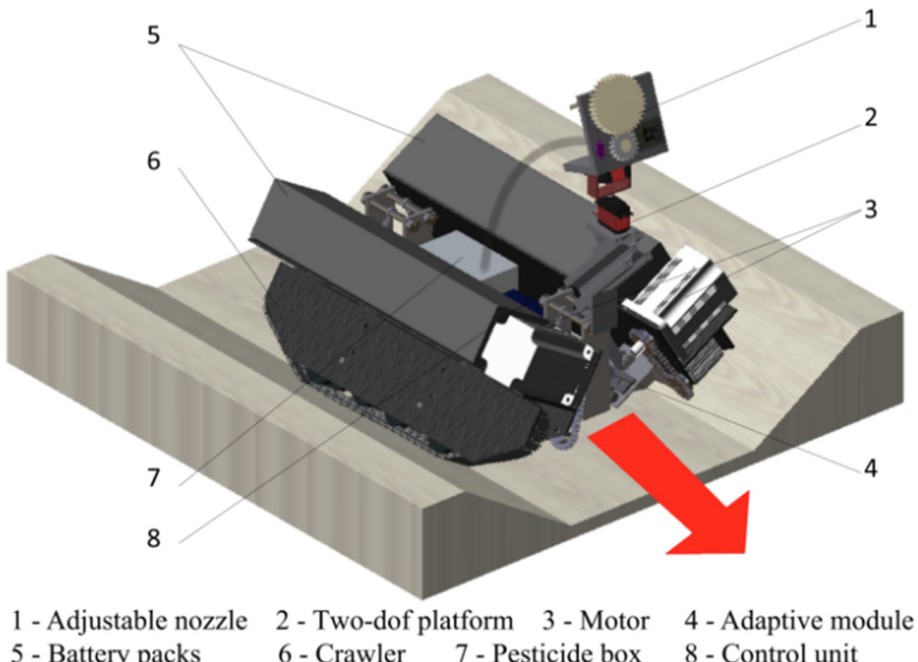

1 - Adjustable nozzle   2 - Two-dof platform   3 - Motor   4 - Adaptive module
5 - Battery packs       6 - Crawler            7 - Pesticide box   8 - Control unit

**Figure 1.** Leaf-bottom pest control robot. The red arrow points to the direction of travel.

The robot autonomously adjusts the target spraying parameters and has the capability to alter the body's shape to adapt to the complex and variable field conditions, as depicted in Figure 2.

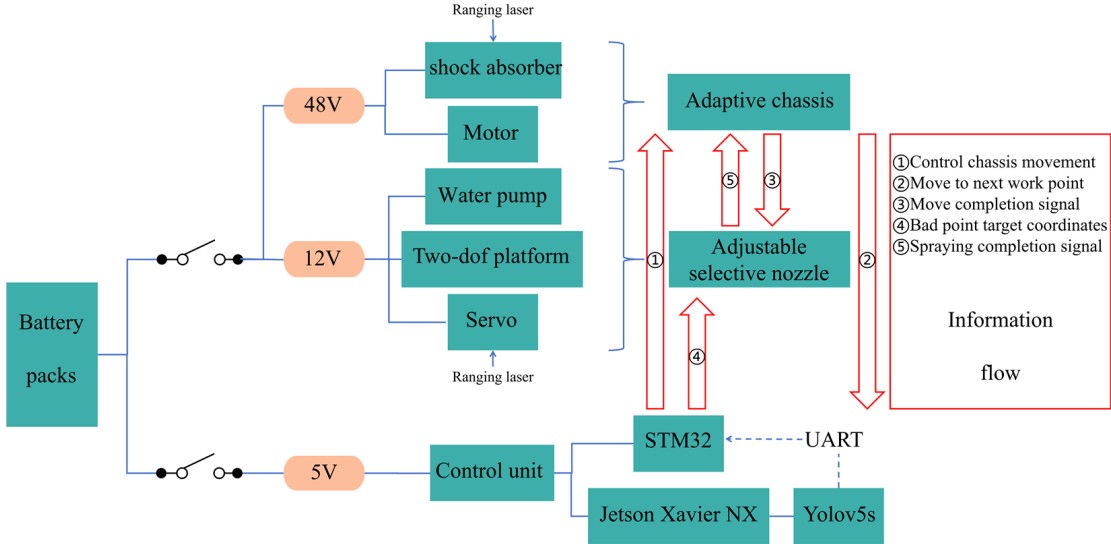

**Figure 2.** Circuit diagram and information flow of the robot.

The chassis is designed based on the MacPherson suspension [34], and the terminal trajectory is modified to change the swing angle of the body with the angle of the ridge slope. The crawler tracks and motors change the swing angles with the main frame in real time to closely match the slope of the ditch. The circuit system is mainly divided into three parts: 5v, 12v and 48v, which, respectively, power the control unit, the selective adjustable nozzle and the adaptive chassis. The information flow during the work process is shown in the red box and red arrow in Figure 2. Firstly, after the machine enters the working point, the onboard camera collects crop images in real-time. Jetson Xavier NX processes these images and transmits environmental conditions back to STM32 through the universal asynchronous receiver and transmitter (UART). Secondly, when the machine

body is stable, the adjustable spraying system starts. The improved lightweight Yolov5s network model identifies the pest gathering area and defective area and calculates the target spraying coordinates. Subsequently, the system combines the data from the ranging laser to determine the target spraying parameters (flow rate, pressure, spray angle) to plan the medication area. The adjustable selective spraying system realizes selective spraying through the two-dof platform and servo and adjusts the dosage through the water pump. It can target the three-dimensional coordinates of pest gathering areas and defective areas according to prevention and control needs, and accurately apply pesticides to the selected locations. The adaptive chassis adapts to different ridge spacing and slopes by changing the swing angle. The adjustable nozzle employs selective spraying technology to achieve effective pest control for crops at different distances and of varying heights. This selective spraying robot is versatile, precise in pesticide application, and minimizes ecological and environmental pollution.

### 2.2. Adaptive Chassis

Traditional agricultural spraying robots typically focus only on the sensors used for target detection and the "robotic arms" used for spray execution. However, a universally adaptive single-ridge chassis, brought about by innovative work forms, is also worthy of attention.

Based on the survey of the pesticide spraying machinery market conducted for this paper, most spraying robots do not possess the capability to adapt to ridges. Farmers often customize agricultural machinery sizes according to the spraying requirements of different crops, which compromises economic efficiency and the utilization rate of machinery. Most machines cannot autonomously adjust their wheel spacing while driving between ridges without reserved driving space, easily damaging the ridge slopes and affecting crop growth. Moreover, due to insufficient contact between the tires and the ground, the body is prone to shaking or even overturning. Indirectly, solutions that reserve space for machinery movement also lead to the wastage of land resources. Thus, it becomes significant for agricultural or spraying machinery to have an automatic adaptive mechanism to deform according to different field ridges. Unlike traditional shock-absorbing mechanisms that can only adapt to a single type of terrain, the adaptive chassis discussed in this article dynamically adjusts the angle between the track drive section and the main frame. The chassis can match different ridge spacings and slopes, as shown in Figure 3.

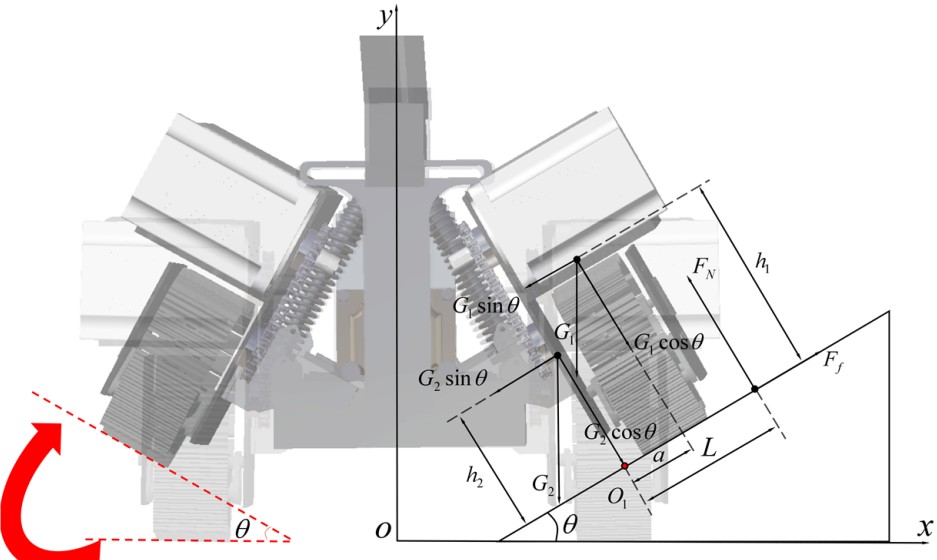

**Figure 3.** The schematic of the chassis deformation and static analysis.

Ridge slope pressure concentration model: The machine is affected by internal and external forces, which will cause uneven stress on the parts contacting the ridge slope. In order to avoid damage to the ridge slope, the more dispersed the supporting force of the ridge slope to the crawler track, the better. We conduct static analysis on the condition of the machine on both sides of the ridge, as shown in Figure 3.

$G_1$ is the gravity on the connection point of the adaptive module. $G_2$ is the gravity on the single-sided drive module. $\theta$ is the ultimate ridge slope angle. $h_1$ is the distance between the centre of gravity of the driving module and the ridge slope. $h_2$ is the distance of the connection point perpendicular to the ridge slope. $a$ is the distance between the centre of gravity of the drive module and the torque point $O_1$ projected onto the ridge slope. Taking $O_1$ as an example, the critical support point of a single-sided track undergoes torque balance analysis, which yields the equation:

$$\sum M_{O_1} = F_N \cdot L + G_1 \cdot sin\theta \cdot h_1 + G_2 \cdot sin\theta \cdot h_2 - G_1 \cdot cos\theta \cdot a \tag{1}$$

From the mechanical analysis, it is known that the vertical supporting force $F_N$ is:

$$F_N = (G_2 + G_1)cos\theta \tag{2}$$

If no stress concentrations occur in a single-sided component on a ridge slope, the moment arm of supporting force $L \geq 0$ must be met, i.e.:

$$L = \frac{G_1 \cdot cos\theta \cdot a - G_1 \cdot sin\theta \cdot h_1 - G_2 \cdot sin\theta \cdot h_2}{(G_2 + G_1)cos\theta} \geq 0 \tag{3}$$

Based on known chassis parameters, the chassis is 0.6 m long and 0.5 m wide, the single-sided module weighs 12 kg, the middle of the body is unladen at 4 kg, and the maximum capacity of the water tank is 8 L. Thus, the maximum ridge slope angle in both loaded and empty states can be calculated. The extreme ridge slope angle can be obtained as follows:

$$\begin{cases} \theta \leq tan^{-1}\left(\frac{4a}{4h_1+h_2}\right), \ fully \ loaded \\ \theta \leq tan^{-1}\left(\frac{12a}{12h_1+h_2}\right), \quad empty \end{cases} \tag{4}$$

From the above Equation (4) and Figure 3, it can be seen that the relationship between the size design of the adaptive chassis and the ultimate ridge slope is obvious. The limit ridge slope allowed by the adaptive chassis is only related to the center of gravity position of the drive module and the connection point of the adaptive module. The minimum limit ridge slope angle occurs when fully loaded. Therefore, the lower the center of gravity of the drive module and the connection point to the adaptive module, the wider the drive module. That is, the smaller the values of $h_1$ and $h_2$. Or the wider the width of the drive module, the larger the value of $a$. This makes the larger the ultimate ridge slope angle, the more uniform the stress distribution on the ridge slope. On the contrary, the smaller the limit ridge slope angle of the adaptive chassis, the more concentrated the stress distribution on the ridge slope.

The adaptive module is designed based on the MacPherson suspension in this paper and is shown in Figure 4. The terminal trajectory is modified to change the swing angle of the body with the angle of the ridge slope. The variation in the suspension travel achieved by installing springs between the driving mechanism and the main frame is utilized. In the case of uneven force, the adaptive module can make the tracks on both sides change along the preset trajectory. Change the swing angle in real-time to make the crawler track completely fit the field ridge. This can increase the ground contact area and reduce the pressure on the field from the driving part.

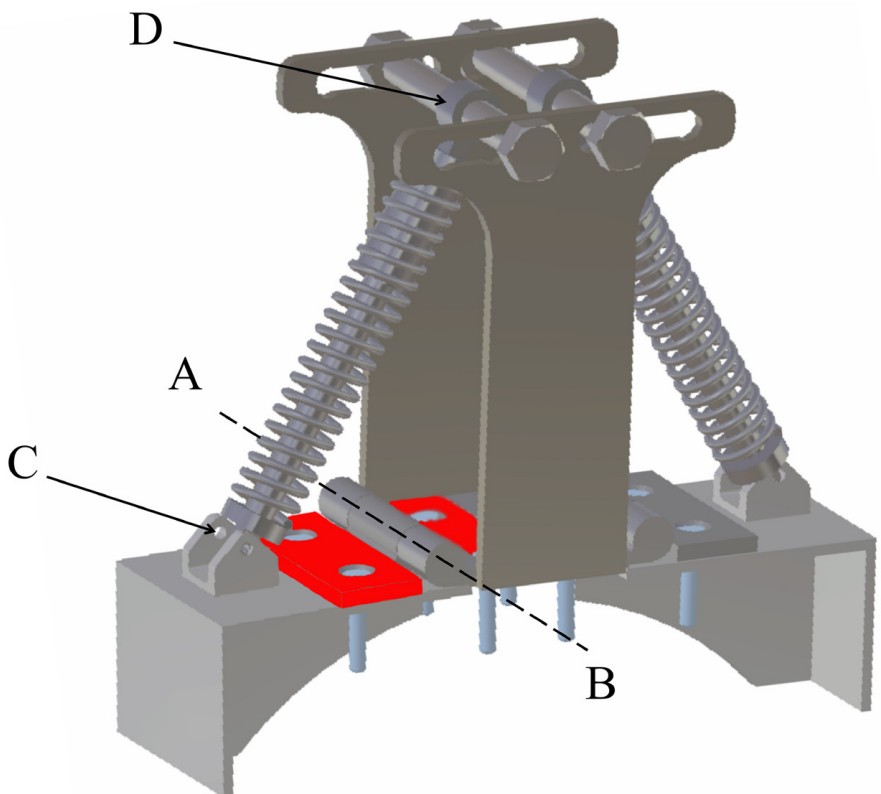

**Figure 4.** Schematic diagram of the adaptive module with reference lines marked. **A, B** is the rotation axis; **C, D** is the lower and upper fulcrum.

The amplitude and speed of angle changes during adaptive chassis deformation are important reference factors. This paper performs a spatial motion analysis on the adaptive module. Discuss how to keep the adaptive module within the appropriate angle change range while having a fast response speed. The following assumptions are made for the model:

(1)  $C'(x_e, y_e, z_e)$ and $D''(x_d, y_d, z_d)$ are the lower and upper fulcrums of shock absorption, respectively.
(2)  AB is the rotation axis.
(3)  The angle of rotation of plane CAB around the $Y$-axis is swing angle.

The rotation matrix:

$$R_y(\theta) = \begin{bmatrix} cos(\theta) & 0 & sin(\theta) \\ 0 & 1 & 0 \\ -sin(\theta) & 0 & cos(\theta) \end{bmatrix} \tag{5}$$

The coordinate transformation of point C can also be obtained:

$$C' = R_y(\theta) \cdot \begin{bmatrix} x_{c0} \\ y_{c0} \\ z_{c0} \end{bmatrix} \tag{6}$$

Then, calculate the distance from $D''$ to $E''$ (The length of DE):

$$CD = \sqrt{\begin{array}{c}(x_d - x_{c0}cos(\theta) - z_{c0}sin(\theta))^2 + (y_d - y_{c0})^2 \\ + (z_d + x_{c0}sin(\theta) - z_{c0}cos(\theta))^2\end{array}} \tag{7}$$

This paper uses MATLAB R2023a and formulas to simulate and analyze the adaptive module (Figure 5). When the initial installation angle of the shock absorber is different, the

swing angle change range (*x*-axis) corresponding to the maximum response speed (slope) also changes. This paper subsequently verifies its correctness through Adams motion simulation analysis (Section 3.1). D1, D2, and D3 represent the initial angle between CD and plane CAB increasing in sequence. It can be seen from the figure that the larger the initial angle, the faster the response speed at low-angle ridge slopes. On the contrary, the smaller the initial angle, the faster the response speed on high-angle ridge slopes.

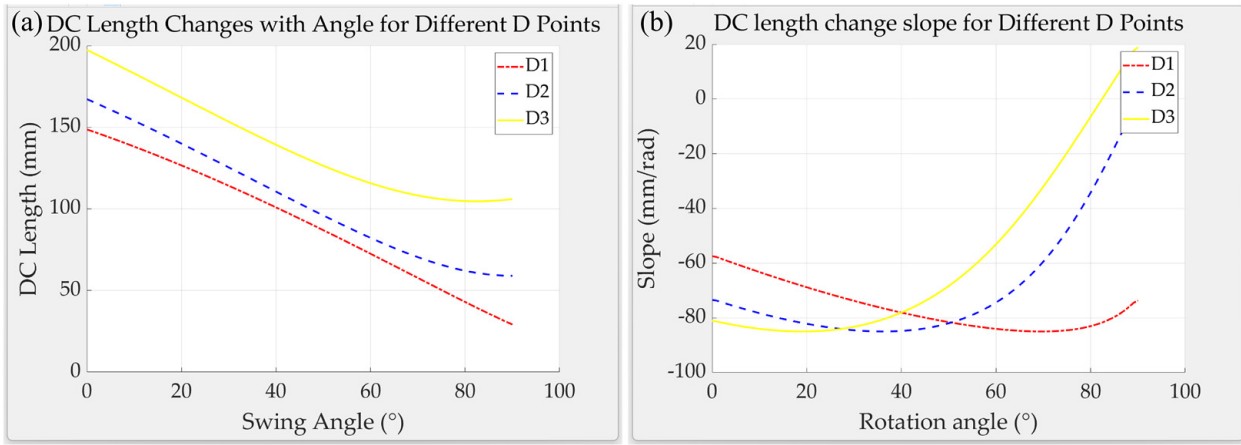

**Figure 5.** (**a**) Curve of shock absorption length changing with swing angle at different installation angles; (**b**) curves of the slope of the shock absorption length change in three cases.

### 2.3. Adjustable Selective Nozzle

In the entire process of pesticide spraying, the level of atomization of the pesticide also affects the effectiveness of the pesticide. Due to the low directivity of atomized particles, most pesticides remain in the air, on the surface of crops, or are dissolved into the plant. The soil reduces the efficacy of the medicinal solution and causes a certain degree of pesticide pollution. Common solutions, such as installing electrostatic generators, can only partially solve air residue issues. Cone-shaped nozzles typically have higher directivity than mist-shaped ones, and their atomization degree is lower. Using visual recognition, laser ranging, point cloud segmentation, and other perception methods combined with array nozzles, height-adjustable mechanical structures, and other methods to improve targeting effects has become a trend. In addition, the downward spraying scheme has limitations. The fact that pests and diseases are concentrated on the underside of leaves inevitably results in most pesticides being attached to the leaf surface. At the same time, in order to achieve precise application of pesticides, the horizontal and vertical directions of the nozzle need to be able to adjust with the position of the bad point.

To address this, this paper designs an upward-spraying adjustable selective nozzle. The nozzle consists of three parts:

(1)  Automatic spray cone angle adjustment module (changes the angle of the spray);
(2)  Feedback diaphragm liquid pump with adjustable pressure (changes the height of the water column);
(3)  Two-dof tracking gimbal carrying the nozzle (for selective spraying).

In the pesticide application pipeline, we designed a set of circulating relief valves that protect the internal pressure of the pipe while using the backflow liquid to continuously stir the liquid in the water tank, ensuring the uniformity of the medication, as shown in Figure 6b.

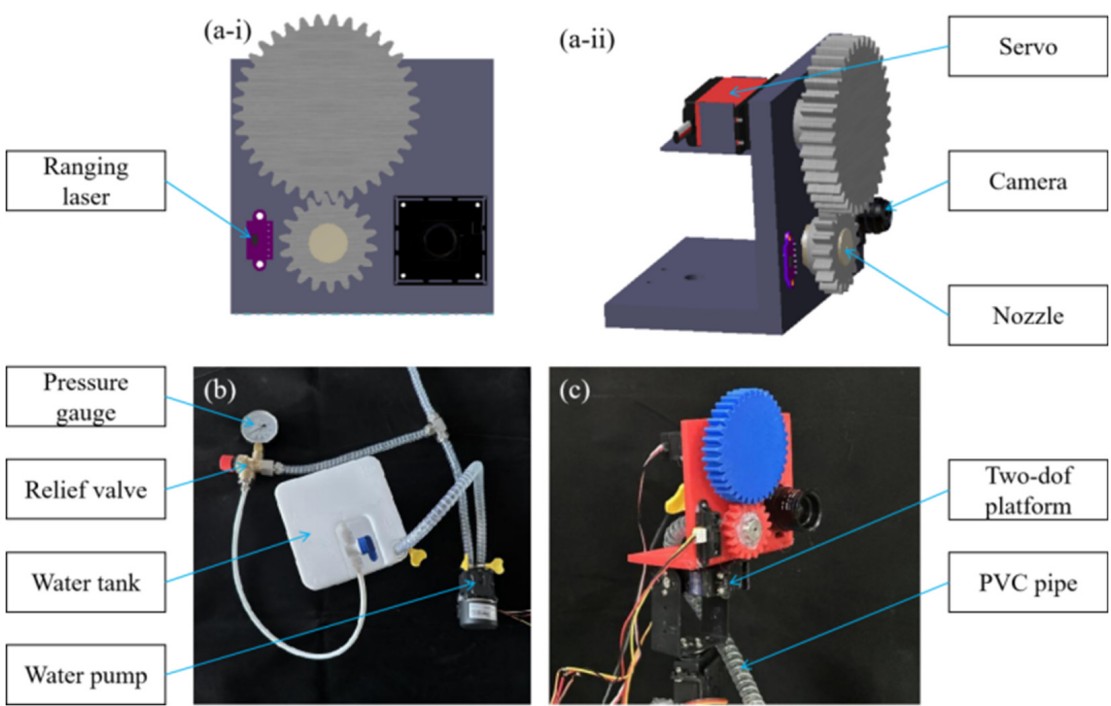

**Figure 6.** The modelling and physical model of the adjustable selective spraying system. (**a-i**) and (**a-ii**) are 3D modeling diagrams; (**b**) is a physical connection diagram of the circulation loop and the water pump; (**c**) is a physical diagram of the selective adjustable nozzle.

Three components form an adjustable selective nozzle, which can not only realize fixed-point spraying but also adjust the spraying area according to the scope of damage caused by diseases and insect pests. Adjust the spraying lift according to crops with different ground clearances. The two-dof pan-tilt locates the defect coordinates in turn according to the recognition results to realize the selective function. The two-dof platform consists of two digital servos (Figure 6c) and is controlled by the PWM signal of STM32. This paper uses the PID (Proportional, Integral, Derivative) fuzzy control method to adjust the PWM signal to reduce the error of the nozzle pointing. A separate servo is used to adjust the spray cone angle of the nozzle (Figure 6a–ii), and a laser radar is used for auxiliary calculation (Figure 6a–i).

As shown in Figure 7, when the coordinates of the pest defect are generated by the multi-CBAM-YOLOv5s, the machine will selectively locate and automatically adjust the spraying after the body is stable. First, when receiving the coordinates, the program will sort the coordinates. Different coordinates correspond to different PWM signals of the three servos, corresponding to the horizontal, vertical and spray cone angles, respectively. The two-dof platform controls the nozzles to point to the target in turn, and continuously runs to form the fitting spray curve path of the pest defect. For closer targets, the spraying system will reduce the water pressure of the diaphragm pump, and reduce the height and distance of the overall water column. For farther targets, the water pressure will increase. In order to concentrate the droplets of the drug mist within the range of the bad points, the pulse number calculated by Formulas (8) and (9) controls the servo to adjust the spray cone angle. The above process is accompanied by laser ranging, and the adjustment of the spraying parameters is controlled according to the formula. When spraying is completed, the machine will go to the next predetermined working point. Based on this liquid spraying system, our equipment can accurately adjust the spraying plan according to the distance and invasion range of the pest gathering area. Such a system can provide accurate and efficient spraying operations, reducing pesticide waste and environmental pollution.

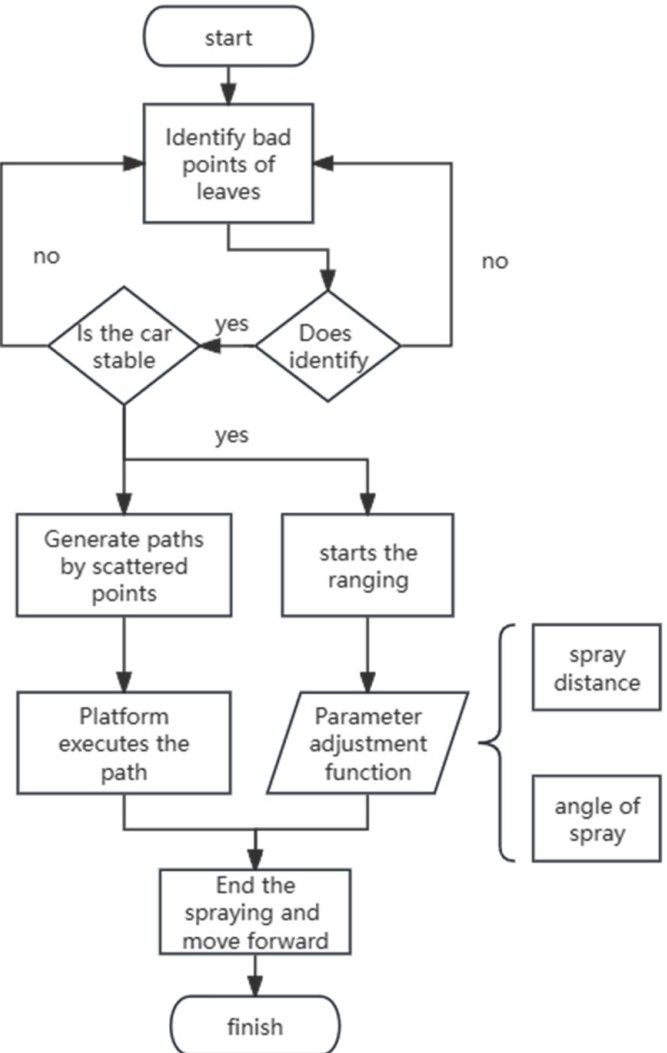

**Figure 7.** The pesticide application flowchart.

According to Figure 8, the rule can be derived that the angle of the spray is proportional to the recognized area of pests and diseases, and inversely proportional to the distance from the nozzle to the leaves. The following assumptions are made for the model:

(1) Considering the anticipated operating scenarios and preliminary experimental results, the plan sets the spray cone angle variation to 10°~60°, corresponding to a controllable servo angle of 270°.

(2) Considering the precision of edge angle control of the servo, the servo angle control is set within a middle range, hence the gear ratio design is 3:1.

The spray angle and servo pulse formula can be obtained as follows:

$$\alpha = 2 \cdot arctan\left( \sqrt{\frac{S}{\pi L^2}} \right) \tag{8}$$

$$Pulse = \frac{(\alpha - 10°) \times 720°}{50° \times 3 \times 270°} \times 2000 + 500 \tag{9}$$

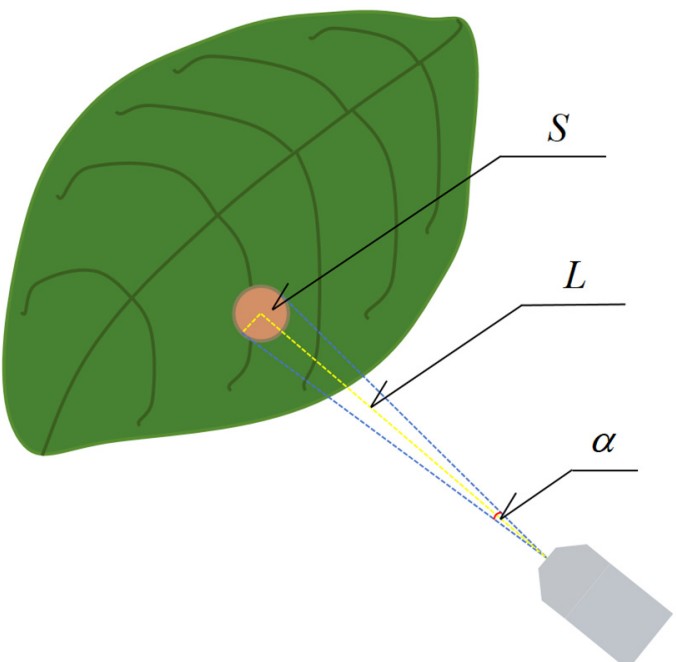

S - Identification area of pests and diseases;
L - Distance from nozzle to leaf;
α - Angle of spray.

**Figure 8.** The parameter adjustment function schematic.

*2.4. Multi-CBAM-YOLOv5s*

In this paper, a network is designed to improve the detection accuracy of spot-type lesions on the underside of leaves of crops such as corn, soybean, and pepper, which is a key driver for the development of pest removal systems. This technology requires the image preprocessing of the collected damaged leaf dataset and input into the neural network model to achieve precise recognition of leaf defects. Small target precise recognition technology can identify tiny leaf-bottom spotted bad point areas in complex and diverse environments, providing technical support for precision agriculture.

In order to eliminate the influence of complex field environments, this article sampled photos of diseases and insect pests on the back sides of leaves of four crops to establish a dataset. The dataset is preprocessed such as binarization, denoising, and morphological opening and closing operations. To achieve higher accuracy and faster detection speeds, this paper improved the multi-CBAM-YOLOv5s recognition network based on YOLOv5s and introduced a CBAM attention mechanism for tiny leaf-bottom spotted bad point identification, as shown in Figure 9.

The network structure of YOLOv5s is mainly divided into four parts: backbone, neck, head and post-processing. To achieve target detection with a few datasets, we selected YOLOv5s as the base network. It can improve information flow and feature expression capability. CBAM is a convolutional block-based attention mechanism presented by S Woo [35] in 2018. It combines the channel attention mechanism and the spatial attention mechanism (Figure 10), which can adapt to the network and significantly improve the correct recognition rate of spotted bad points.

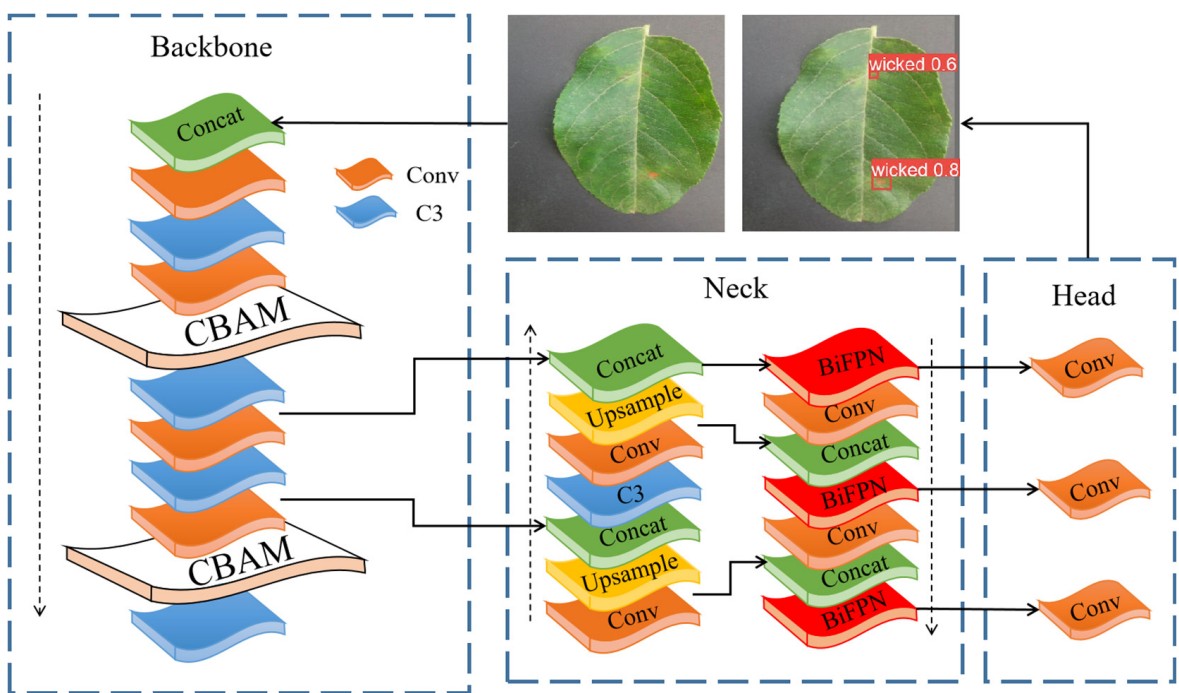

**Figure 9.** The multi-CBAM-YOLOv5s structure.

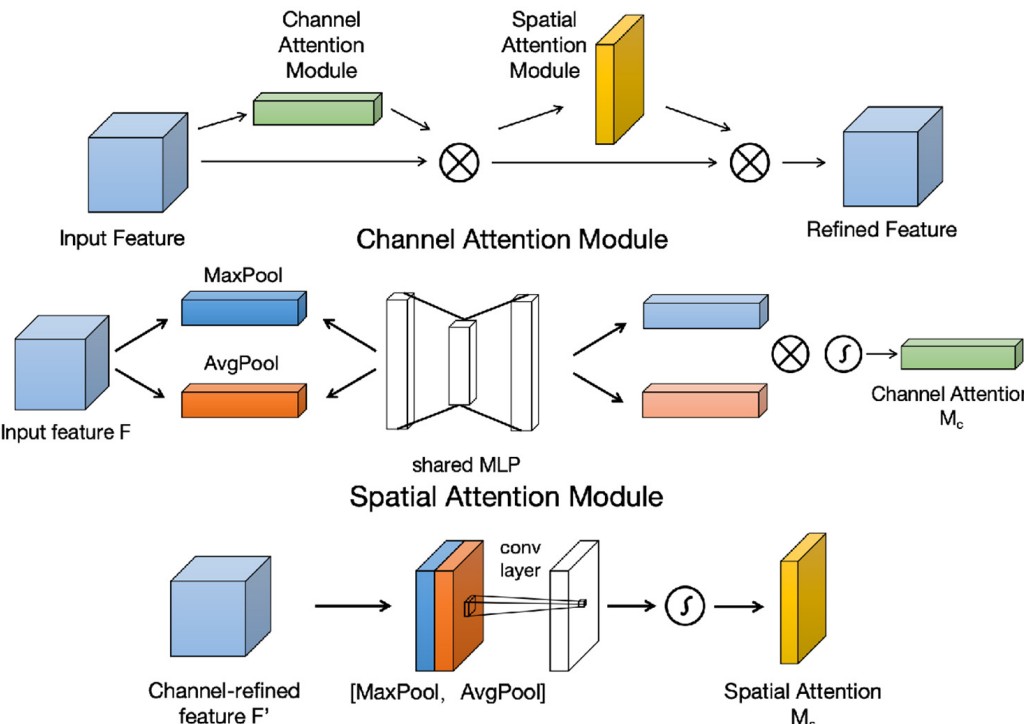

**Figure 10.** CBAM structure: channel attention mechanism and spatial attention mechanism.

In the backbone framework, backbone networks are usually used as feature extractors. By extracting and analyzing the features of the target detection object, this paper inserts a CBAM attention mechanism module between the Conv layer (Convolutional Layer) and the C3 layer (CSPC3 Layer) in the backbone framework.

Inserting the CBAM module after the Conv layer allows the network to pay more attention to important channel information while extracting features. This part reduces attention to redundant information and enhances the efficiency of feature expression.

Inserting the CBAM module before the C3 layer allows the network to better focus on the important information in the feature map. This part reduces missed detections and false detections and enhances the accuracy of target detection.

The image to be detected is input into the network, and after convolution by the Focus module, a $320 \times 320 \times 32$ feature map is output. This feature map, after passing through the CAM and SAM channel attention modules, re-enters the lightweight convolutional neural network CSPDarknet53 (C3). The PANet (Path Aggregation Network) is used as its neck network. By fusing features of multiple scales and performing up-sampling operations, the accuracy of target detection is improved. In the head network framework, the target detection prediction boxes and category probabilities are generated.

This article uses image preprocessing algorithms to improve leaf image quality and highlight leaf characteristics and their pests and diseases (Figure 11). The original color leaf image is converted into grayscale images, these images are then denoised. Median filters effectively remove noise while keeping image edges clear. In order to enhance the distinction between leaves, pests and diseases, a color space conversion algorithm is used to convert the image into an HSV color space. Threshold segmentation is used to extract green leaves and brown diseases and insect pests. Features of leaves, pests and diseases are extracted through a series of image preprocessing algorithms and technologies such as grayscale processing, denoising, adaptive threshold binarization, morphological operations, and color space conversion, enhancing its discrimination and laying the foundation for subsequent extraction of leaf, disease and insect pest characteristics.

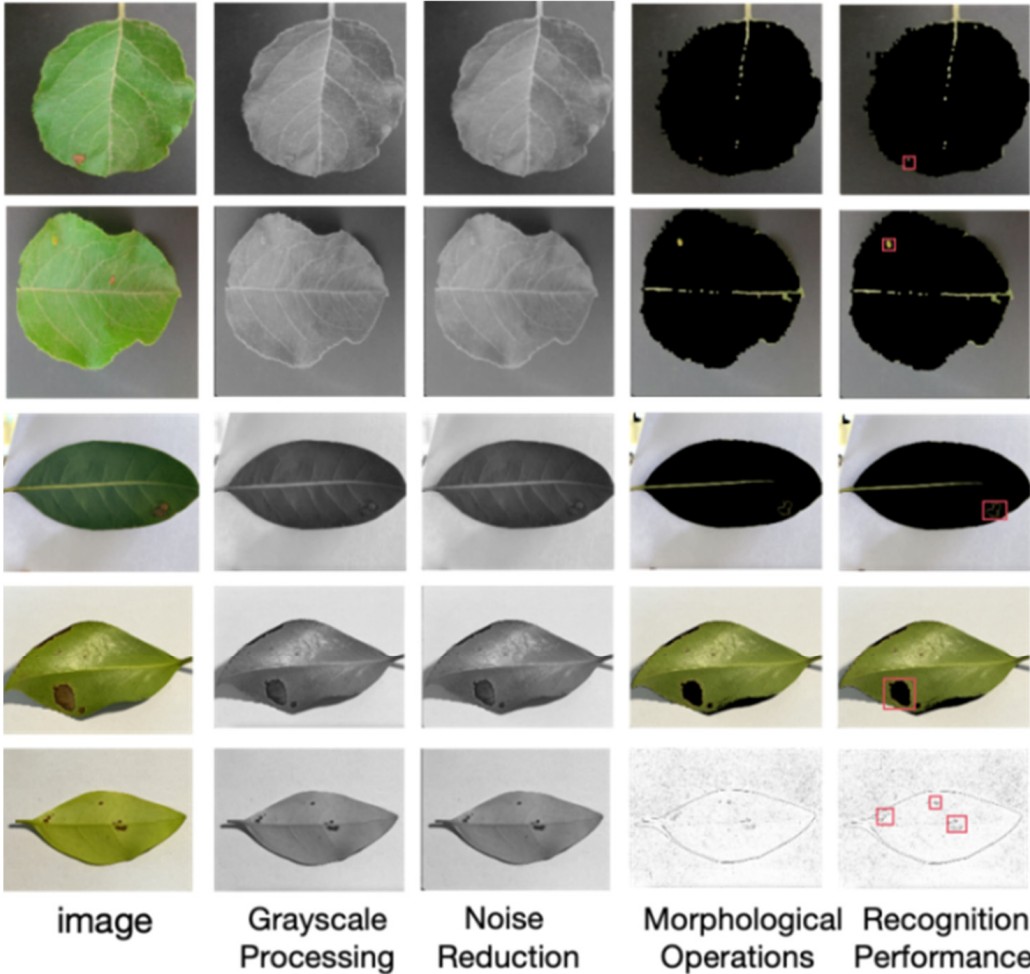

**Figure 11.** Image preprocessing. The red box is the recognition box.

In order to enhance the extent of the neural network's focus on detecting objects, this paper adds attention mechanisms to the network. The attention mechanisms in deep learning are similar to those in human vision; they concentrate on important points amidst a plethora of information, selecting critical information and ignoring unimportant details. This paper selects three attention mechanisms: SimAM, SE, and CBAM. SimAM is a lightweight, parameter-free convolutional neural network attention mechanism, which generates attention weights by calculating the local self-similarity of feature maps. SE is termed the Squeeze-and-Excitation (SE) block, with the goal of improving the quality of representations produced by a network by explicitly modelling the interdependencies between the channels of its convolutional features. CBAM combines the output features of the channel attention module and the spatial attention module through element-wise multiplication to obtain the final attention-enhanced features.

The three attention mechanisms were added to the base network and trained for 40 epochs, and the accuracy when setting the intersection over union (IoU) was used as the evaluation indicator. Based on the training results, the accuracy, mAP_0.5 (average accuracy when the IoU threshold is 0.5) and mAP_0.5:0.95 (IoU threshold is in the range of 0.5:0.95, step size is 0.05, average accuracy) were compared. It was found that the network with the CBAM attention mechanism has the highest recognition accuracy than the other two networks (Table 1). The value of mAP_0.5:0.95 verifies that the model not only has good predictions in general coverage areas (such as IoU threshold is 0.5) but also maintains good accuracy under more stringent matching conditions (such as IoU threshold is 0.95). This is a key indicator of the leaf-bottom spot bad point detection network model.

**Table 1.** Comparison of CBAM module with other modules.

| Deep Learning Models | Precision | Recall | mAP_0.5 | mAP_0.5:0.95 |
|---|---|---|---|---|
| Yolov5s-SE | 0.959 | 0.942 | 0.958 | 0.47 |
| Yolov5s-SimAM | 0.973 | 0.919 | 0.954 | 0.52 |
| Yolov5s-cbam | 0.988 | 0.956 | 0.990 | 0.58 |

Figure 12 shows the differences between the improved network and the base network. From 2-a and 2-b, it can be observed that after adding the CBAM attention mechanism detection module, the curve's highest point is noticeably closer to the upper right corner, indicating that the model can simultaneously ensure high precision and high recall during prediction, i.e., the prediction results are more accurate. From 2-b and 2-c, it can be observed that increasing the number of training rounds also can enhance the prediction.

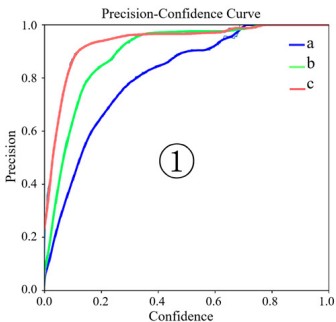 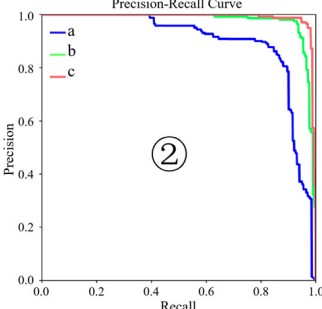 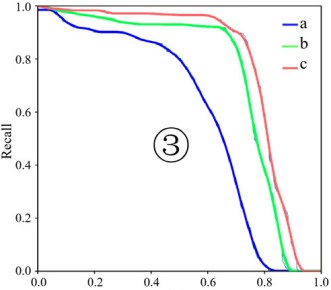 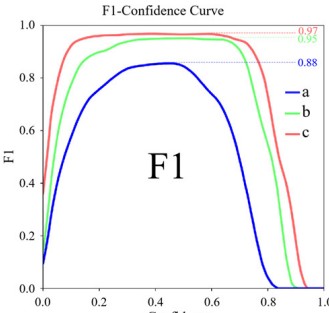

**Figure 12.** Comparison between multi-CBAM-YOLOv5s and basic network model. Model a is Basic-YOLOv5s trained for 40 rounds; Model b is multi-CBAM-YOLOv5s trained for 40 rounds; Model c is multi-CBAM-YOLOv5s trained for 100 rounds. **1** is the Precision-Confidence curve; **2** is the Precision-Recall curve; **3** is the Recall-Confidence curve; the curve of **F1** and confidence represents the comprehensive score.

The F1 score is a measure of classification, being the harmonic mean of precision and recall, with its peak value indicating better performance. When the confidence threshold is raised, it can filter out most false targets, making the category detection accurate, i.e., high precision. However, when the training rounds for the dataset are increased, the F1 score gradually approaches one, to avoid training results becoming overfitted, this paper sets the dataset training rounds to 100 rounds. On this basis, this paper introduces a CBAM attention mechanism module, which can raise the F1 score to 0.97, achieving normalized classification data.

## 3. Experimental Verification

To verify the effectiveness of the design scheme and the functionality of the prototype, this section sets up tests for the chassis movement function and the precision spraying system. In the chassis movement function test, this paper utilizes Adams to simulate the adaptive chassis in a 0° to 35° simulated field environment, monitoring the angle variation of the adaptive module to assess its performance. At the same time, this article modified the initial angle of the model suspension and conducted motion simulation analysis. The bouncing behavior of suspensions with different initial shock absorption angles when entering a 30° ridge slope was tested. Simultaneously, a simple physical model and site are constructed to conduct physical tests on the adaptive chassis under three working conditions: on flat ground, bilateral up-ridge (15°), and single-sided up-ridge (30°). In the precision spraying system test, this paper conducts recognition tests on different datasets of leaf leaf-bottom spotted bad points to preliminarily verify the network model. Builds a complex field environment to simulate corn, pepper, and soybean crops. Manually arranges test defect points according to common distribution areas of leaf diseases and insect pests. Tests the identification of leaf-bottom spotted bad point detection and precise spraying functions under complex conditions.

### 3.1. Adaptive Chassis Movement Function Test

This article conducted a motion simulation test of the adaptive chassis in Adams (Figure 13a). Two common scenarios were tested: from flat ground to furrow (0° to 30°) and on-furrow angle change (30° to 35°). These tests are designed to simulate the operating conditions of the adaptive chassis in the field. At the same time, the angle of the hinge connecting the main frame and the drive module is monitored. The effectiveness of the adaptive chassis is evaluated by the degree of change in the curve. The results are shown in Figure 14. This shows its adaptability to various ridge slopes. Figure 13b shows the real thing running on a simulated single ridge built with wooden boards. Figure 13c is a physical picture of the complete machine.

In Figure 14, the red curve is the change of the swing angle of the adaptive chassis. The blue curve is the change in furrow angle (based on the front angle). From this analysis, when the slope is switched, the adaptive chassis also completes the angle change in time. When the slope is stable, the swing angle of the adaptive chassis is stable. The angle difference between the red curve and the blue dashed line is a reasonable systematic error (about 5° due to flexible deformation). Observe the slope change: as the ridge slope angle increases, the red curve fluctuates upward. When the ridge slope is switched, the shaking of the adaptive chassis causes uneven stress on the deformation mechanism. This allows the adaptive chassis' swing angle curve to climb quickly. In general, the adaptive chassis responds quickly to changes in the slope of the field ridge, enabling real-time adaptation to different angles of the ridge slope. The versatility of complex field work tools was greatly enhanced, providing new configurations and new solutions for micro precision agricultural machinery.

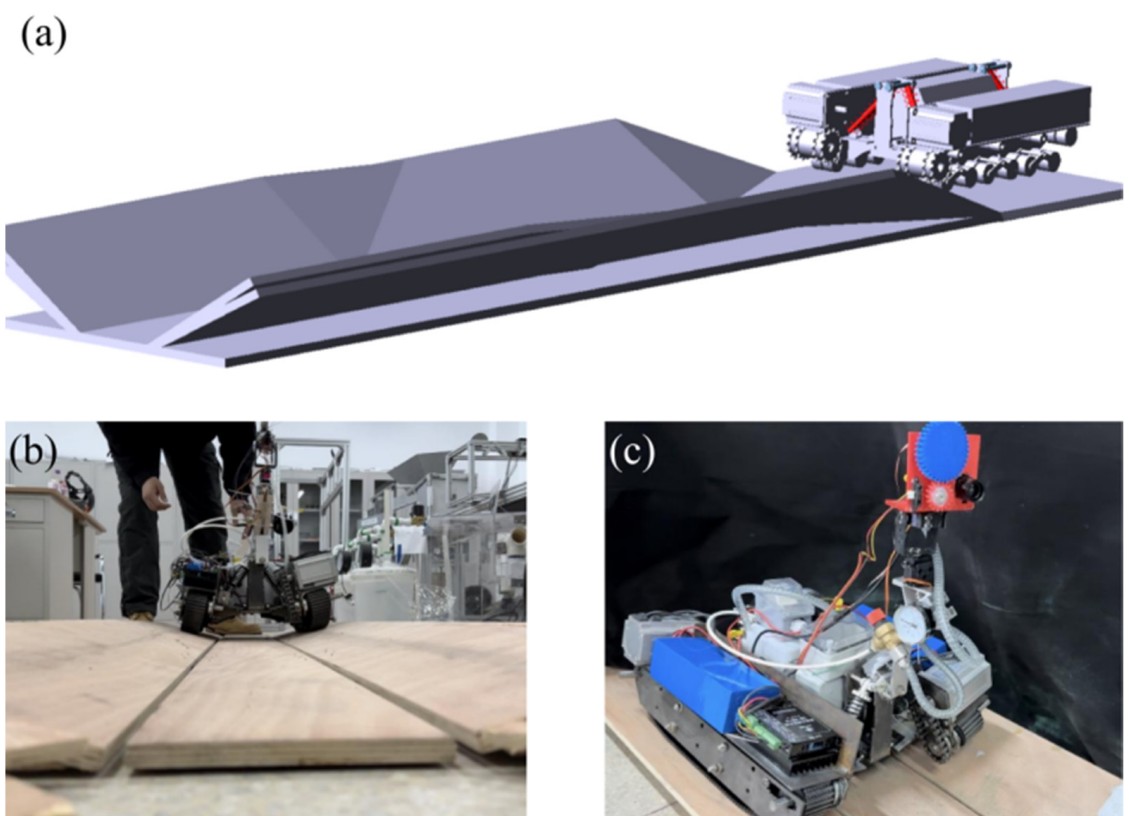

**Figure 13.** The Adams motion simulation of the adaptive chassis. (**a**) is a traveling simulation test in Adams; (**b**) is a real machine simulation test in the laboratory; (**c**) is a static schematic diagram of the machine on the ridge slope.

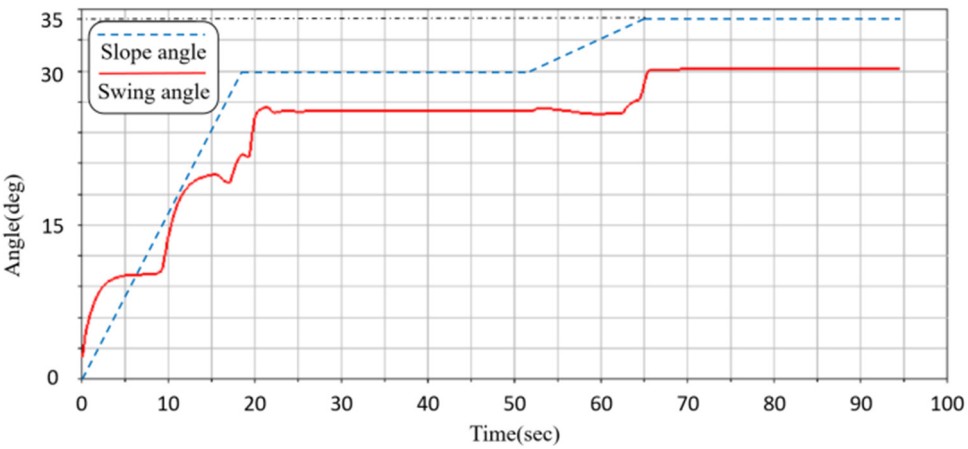

**Figure 14.** Adaptive chassis swing angle and slope angle change curve diagram.

In order to verify the correctness of the adaptive deformation mechanism, this paper conducts motion simulations of shock absorption at different initial angles (Figure 15). The experiment follows the following assumptions: (1) the initial position of the upper fulcrum of the shock absorber is used as a variable; (2) the shock absorber damping and pre-compression are consistent; (3) the vehicle weight is consistent and the speed is 1m/s; (4) the swing angle is based on the angle change at the front wheel hinge; and (5) he ridge slope angle is 30°.

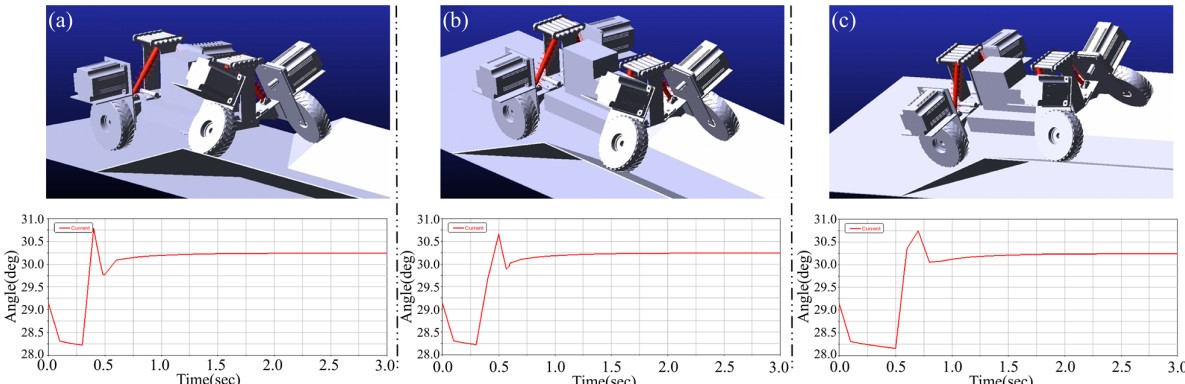

**Figure 15.** Initial angle of shock absorbers and response curves during deformation. Sub-figures (**a**–**c**) are from the Adams simulation test. Different initial installation angles correspond to different response speeds. The curves correspond to the Adams screenshots.

The initial damping angles of Figure 15a–c change by changing the initial position of the upper fulcrum and gradually increase. Observing the changes in the curve from 0.25 s to 1.25 s, it is not difficult to find that the bounce suppression (i.e., the response speed) is proportional to the initial angle. When the initial damping angle increases, the bounce curve becomes flatter, and the peak value becomes smaller. This conclusion can provide a theoretical basis for this type of adaptive chassis and have an important impact on its structural optimization.

This paper constructs a prototype using cast iron, weighing 28 kg when unladen, and uses wooden boards to build a simple variable-angle ridge slope. The adaptive chassis is physically simulated tested separately on flat ground, bilateral up-ridge (15°), and single-sided up-ridge (30°), three working conditions (Figure 16a–c). Observing the actual working conditions in the field (Figure 16d,e), the track fits the ridge slope firmly. The adaptive chassis can achieve conformity with the adaptive angle variation of the ridge slope, reducing damage to the ridge slope, and enhancing the passability of the agricultural machinery.

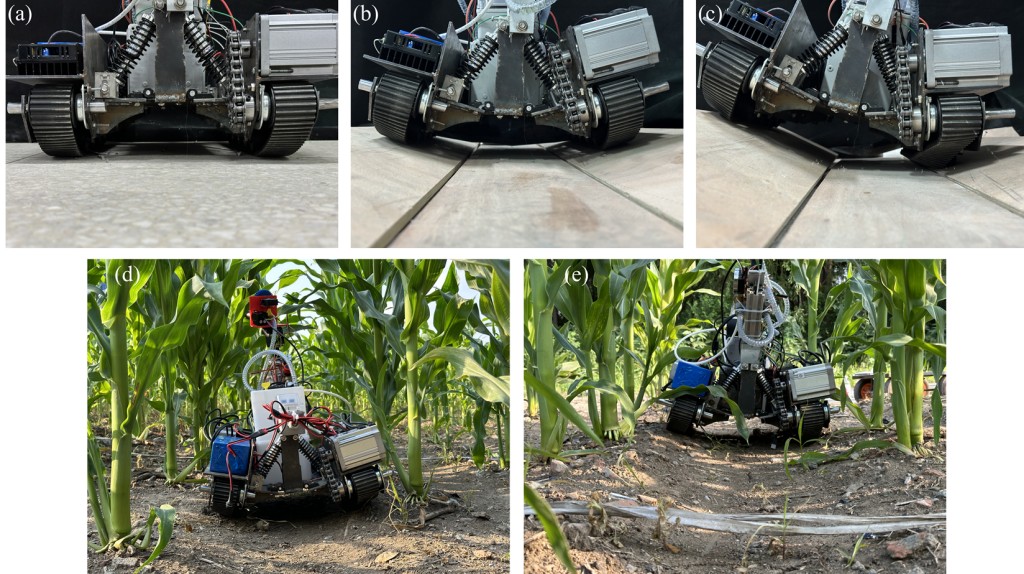

**Figure 16.** The physical test of the adaptive chassis. (**a**–**c**) are the chassis conditions of the actual machine under different working conditions. (**d**,**e**) are the chassis working conditions of the actual machine in the test field.

### 3.2. Leaf-Bottom Spotted Bad Point Recognition and Spraying System Test

In order to verify the feasibility of multi-CBAM-YOLOv5, this paper uses a network model to identify leaf damage of different types and degrees based on a self-built dataset (Figure 17a) and obtains an 85% recognition rate (confidence $\geq 0.6$). At the same time, we conducted field tests in corn test fields (Figure 17b) and found that the model can take and mark out both bad spots on corn and pests attached to leaves (see the appendix video for more details of Supplementary Materials). This test reflects the generalization ability of the network for leaf-bottom spotted bad point detection well.

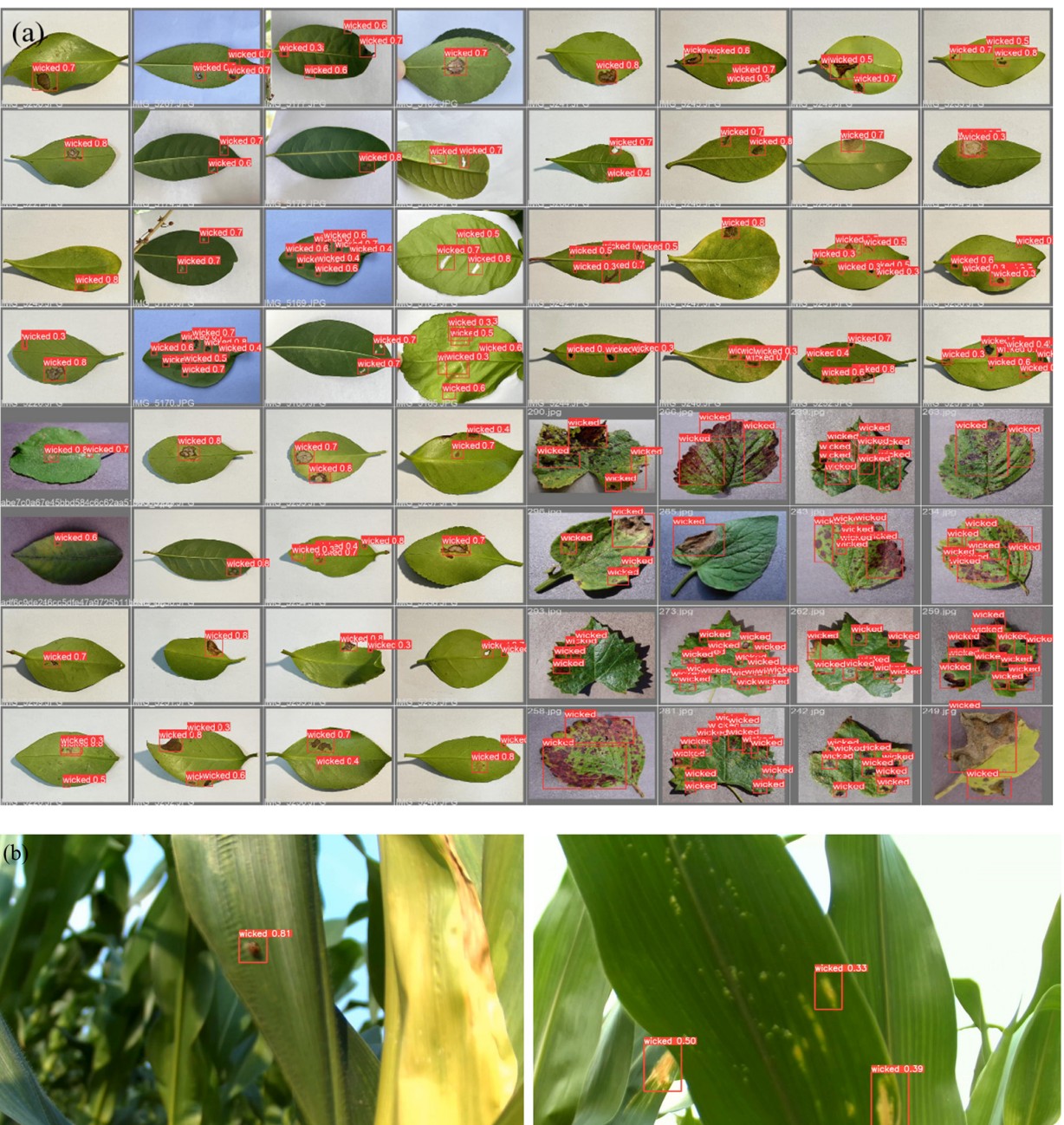

**Figure 17.** The recognition results of the leaf database defect points and field test. (**a**) is the recognition situation of the image stream; (**b**) is a video screenshot of the real-time recognition of the field video stream.

Due to the uncontrollable field environment, it is impossible to reflect the multiple advantages and disadvantages of the network models at the same time. In order to further

verify the generalization ability of the proposed network for spot-type bad spots, this paper simulates the distribution area of leaf diseases and pests by artificially arranging bad spots. Use simulated crops to simulate three crops, corn, pepper, and soybean, to simulate a complex field environment (Figure 18). This test can truly reflect the field operations of different crops in a complex environment and is used to verify the generalization ability of the network for spotted bad point detection. As shown below: Figure 18a shows the original image simulating a complex environment, and defect points are artificially arranged in the crop. Figure 18b shows the binary image, clearly showing the spatial distribution of defect points. Figure 18c shows the recognition situation of multi-CBAM-YOLOv5s. It shows that the network can accurately identify defect points in complex environments. When the confidence $\geq$ 0.6, the recognition rate is 77.8%, and there are no misidentifications. The recognition rate is slightly lower than the leaf recognition rate and is mainly affected by factors such as leaf overlap, crop gaps, and small target size. Figure 18d shows the identification of the basic network. The recognition rate is only 11.1% (confidence $\geq$ 0.6). Moreover, gaps between leaves are often misidentified as defect points. The enlarged parts of Figure 18c,d are the same recognition target. The target accuracy of Figure 18c is 0.85, while that of Figure 18d is only 0.66 and is misidentified as two target points.

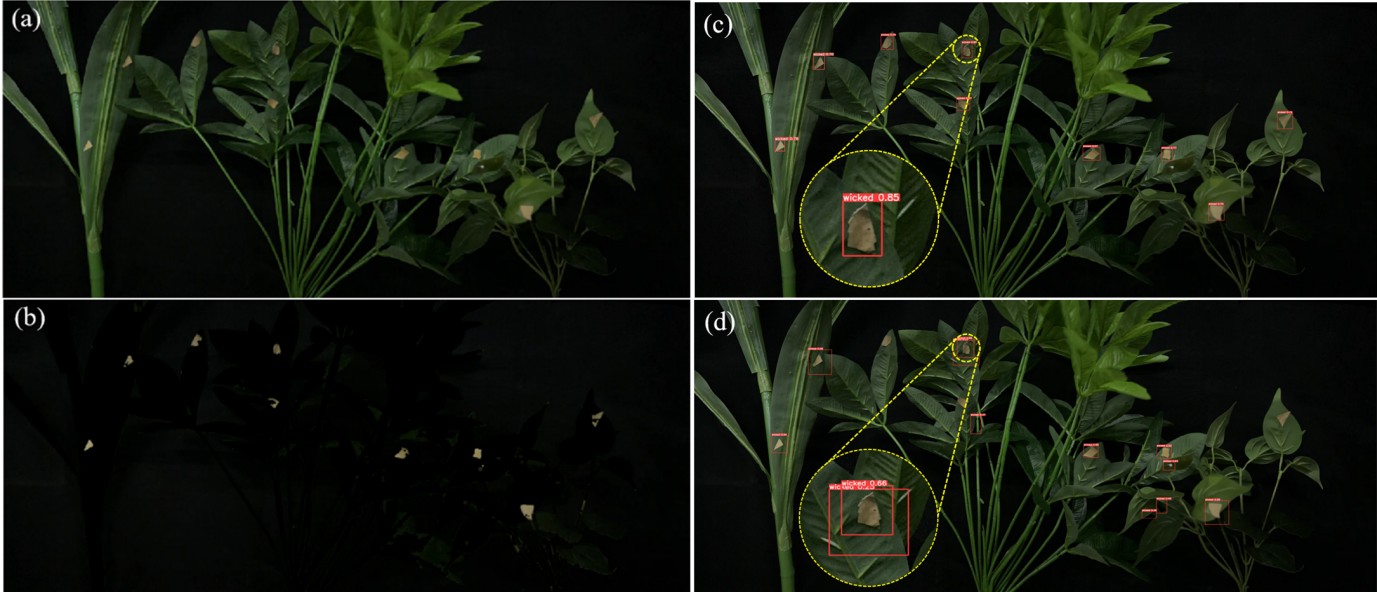

**Figure 18.** The recognition results of the simulated complex environment. (**a**) is the original picture; (**b**) is the image binarization; (**c**) is the recognition situation of multi-CBAM-YOLOv5s; (**d**) is the recognition situation of the basic network.

We manually arranged three identification points representing different heights and distances. Figure 19a tests the adjustable selective spraying effect on targets at different distances and angles. It can be observed from the spray deposition area that the identification point is completely covered, verifying the reliability of Formulas (8) and (9).

In order to simulate the complex field environment mentioned above, this article also conducted a spraying test (Figure 19b). Multi-CBAM-YOLOv5s first transmits the coordinates of the identified defect points to STM32 through the serial port. Then, the coordinates are sorted from left to right and from high to low and fitted into a trajectory for trace spraying. At the same time, the spray system automatically adjusts the spray cone angle by combining the received identified defect point area and real-time laser ranging data. This system increases the rate of pesticide application and reduces the rate of pesticide waste.

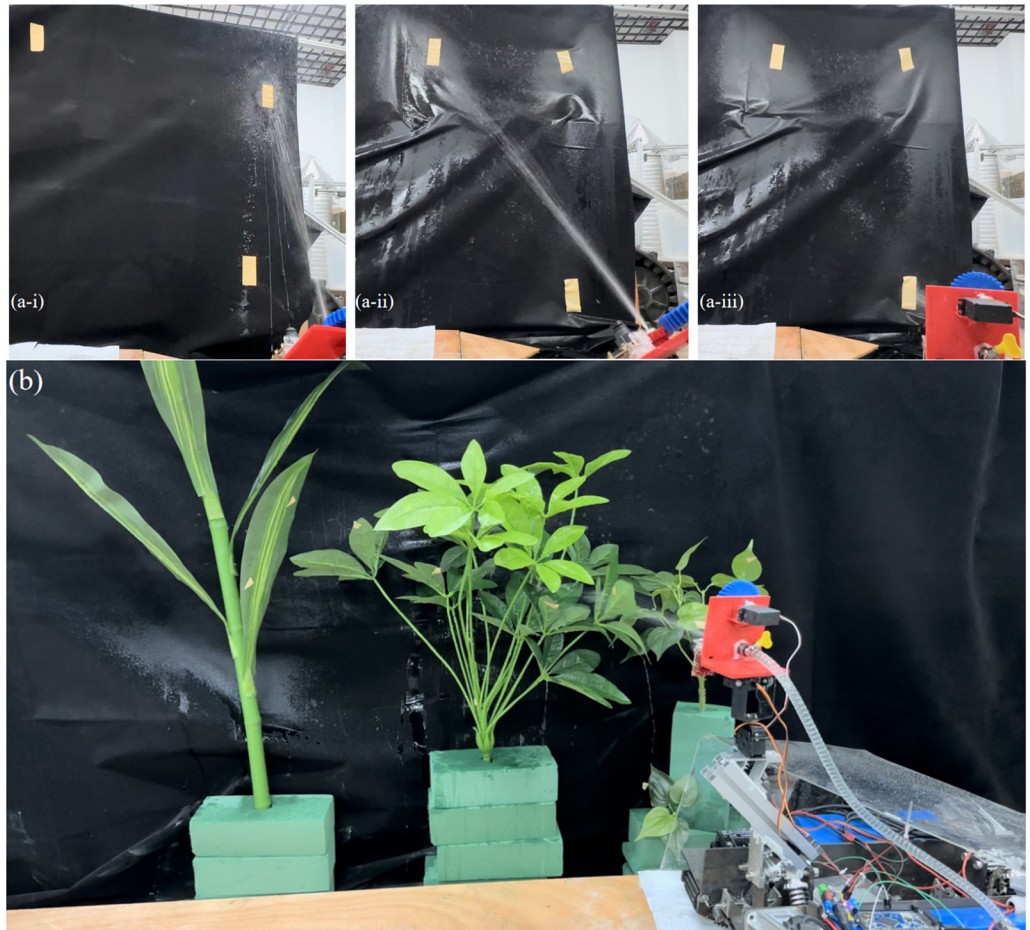

**Figure 19.** The real-life spraying tests. Sub-image (**a**) shows the nozzle's adjustment of spray parameters; (**b**) is a screenshot of the actual spraying.

## 4. Discussion

Traditional spraying agricultural machinery usually has the following problems:

(1)    Spraying the leaf surface, while pests and diseases are usually located at the bottom of the leaf.
(2)    The size is too large for leaf-bottom spraying.
(3)    Different crops require customized machines.
(4)    Non-intelligent operations.

In view of the above problems, this article designs a robot. In the case of a single machine operation, the following can be realized:

(1)    Without damaging the ridge slope, it is possible to enter ridges of different sizes to carry out insect removal operations at the bottom of leaves.
(2)    A variety of pests and diseases can be identified in a complex field environment.
(3)    Automatically adjust parameters to accurately control insect pests on the bottom of leaves.

This paper explores a new method for automatically adapting to ridge slope angles. How to adapt to such terrain challenges is addressed. Previous robots were unable to enter narrow furrows without reserved work space. The adaptive chassis designed in this article can automatically adapt to different ridge and furrow slopes. However, there are still some problems: the car body is easy to shake during the deformation process, the track fit is insufficient, and the contact surface pressure cannot be measured. Future research should focus on optimizing the design by adding stress plates to the shock absorbers and conducting dynamic analysis of the vehicle body.

This article designs an adjustable selective spraying system for tiny targets. In the actual test, the nozzle angle change was not very linear, and the control accuracy was poor. The water jet was severely obstructed by leaves during spraying. Currently, there is a lack of datasets on diseases and insect pests on the undersides of various crop leaves, and the identification accuracy in complex environments needs to be improved. Since it is currently in the prototype design and experimental optimization stage, the single-time endurance (spraying amount and walking distance) and autonomous navigation capabilities are weak. Future research should include redesigning the structure of the nozzle to make its changes more uniform. Adding multiple selective nozzles will solve the inefficiencies and clogging issues of a single angle. More field data collection should be conducted to address the problem of insufficient datasets. The autonomous navigation capability and single-time endurance should be improved around the water tank capacity, power, power consumption and decision-making ability.

## 5. Conclusions

In order to verify the proposed solution, this paper designed an adaptive module and constructed a simulated complex field environment. A prototype robot was built for testing and experimentation of the spray system. In the sports function test, the angular adaptability of the chassis was verified through simulation and physical operation. Adaptive adjustment from $0°$ to $35°$ was achieved, confirming the effectiveness of the design. In the experimental test of the spraying system, the network was trained and tested based on the self-built dataset, and a disease recognition rate of 85% was achieved. In order to further verify the network's generalized recognition ability of leaf-bottom spotted bad point detection, a complex field environment simulating crops was constructed. In the recognition test, a recognition accuracy of 77.8% was achieved, which improved the accuracy compared to the baseline network algorithm. In the simulated spray test scenario, the linkage between the adjustable selective nozzle and the vision system is verified. The spraying system automatically fits the spraying trajectory and executes it accurately, achieving the expected automatic, precise and selective spraying. In the future, we will continue to optimize and improve the chassis and spraying system. We will further verify the actual deployment effect of the robot system and explore intelligent multi-machine linkage and cluster operations.

**Supplementary Materials:** The following supporting information can be downloaded at: https://www.mdpi.com/article/10.3390/agriculture14081341/s1, Video S1: Leaf-bottom pest control robot.

**Author Contributions:** Conceptualization, D.L.; methodology, D.L.; software, D.L., F.G. and Z.L.; validation, D.L., F.G., Z.L., Y.Z. and C.G.; formal analysis, D.L.; investigation, D.L. and F.G.; resources, D.L. and F.G.; data curation, D.L. and F.G.; writing—original draft preparation, D.L. and F.G.; writing—review and editing, D.L. and H.L.; visualization, D.L., F.G. and Y.Z.; supervision, D.L.; project administration, D.L.; funding acquisition, H.L. All authors have read and agreed to the published version of the manuscript.

**Funding:** This work was supported by the Heilongjiang Provincial Natural Science Foundation Project (LH2023C031).

**Institutional Review Board Statement:** Not applicable.

**Data Availability Statement:** Data are contained within the article.

**Acknowledgments:** The authors thank the editor and anonymous reviewers for providing helpful suggestions for improving the quality of this manuscript.

**Conflicts of Interest:** The authors declare no conflicts of interest.

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
