# Peer review of "Design of a Leaf-Bottom Pest Control Robot with Adaptive Chassis and Adjustable Selective Nozzle"

_agriculture, doi:10.3390/agriculture14081341_

Round 1

Reviewer 1 Report

Comments and Suggestions for Authors

This is a paper on the design and testing of a pesticide spraying robot, which discusses the chassis design, nozzle design, and machine vision from three aspects. After reading the entire paper, the reviewer believes that it lacks practical value and has many defects. For the content of the article, the suggestions are as follows:

1. In the abstract, add the design content of the machine.

2. In the introduction section, it is recommended to reduce the number of paragraphs and merge the research background (paragraphs 1-3) into one paragraph and the domestic and international research status (paragraphs 4-7) into 1-2 paragraphs.

3. In the chassis design part, the reviewer is not concerned with the stress analysis of the tracks because the weight of the entire chassis is not heavy. If the author really wants to reflect the workload, it is suggested to focus on navigation, for example: achieving autonomous walking in corn fields.

4. Regarding the nozzle design, the reviewer is unclear about the design principle of the 2-DOF platform. Compared to the adjustment of the nozzle spray cone angle and pressure, the reviewer is more concerned with the macroscopic structural design/control of the nozzle platform for the overall spray direction/angle and spray height. Additionally, how does the author solve the problem of the pesticide tank capacity?

5. In terms of deep learning, the author’s target object introduction is unclear. What kind of plant and pest is the author researching? The reviewer does not believe that the author has trained a universal model because it is not possible. There are many improvements to the Yolo series algorithms on online deep learning websites, and the key is innovation in application.

6. Throughout the entire paper, the author wants to present their work to the readers and explain how their machine was developed and tested. However, the three design parts are not clearly introduced, and the reviewer feels that the author is not addressing the main points.

Comments on the Quality of English Language

None.

Reviewer 2 Report

Comments and Suggestions for Authors

Reviewer 3 Report

Comments and Suggestions for Authors

Dear authors, I appreciate your hard work, and I have the following suggestions:

1.        In the abstract, highlight the key characteristics of the work. You include certain sentences without logical linkage, and you repeat the title. Please rework the abstract and avoid acronyms.

2.        Provide explanations for any acronyms used in text.

3.        Include a figure that explains the relationship between the systems and the purpose of each subsystem. Maybe you can adjust Figure 2, which is a little perplexing because it is based on power feed rather than information flow. Please split the figure 4 into 3 different figures. Please describe in detail the simulations: purpose, methodology, results interpretation.

4.        In figure 5, you can use only the figures b and c. The function of figures a-1 and a-ii is unclear. Please describe in the essay the importance of 3D modeling.

5.        Explain why you used parameter 5 and compare it to other parameters.

6.        Please include information regarding the CBAM attention mechanism module.

Round 2

Reviewer 1 Report

Comments and Suggestions for Authors

None.

Reviewer 2 Report

Comments and Suggestions for Authors

All the Comments are answered well and can be consideration for publication.